# Outcome of Non-Muscle Invasive Upper Tract Urothelial Carcinoma Receiving Endoscopic Ablation: An Inverse Probability of Treatment Weighting Analysis

**DOI:** 10.3390/jcm11051307

**Published:** 2022-02-27

**Authors:** Chih-Yu Shen, Yeong-Chin Jou, Wei-Chih Kan, Tzong-Shin Tzai, Yuh-Shyan Tsai

**Affiliations:** 1Department of Urology, Tainan Hospital, Ministry of Health and Welfare, Tainan City 70043, Taiwan; cyshen780501@hotmail.com; 2Department of Urology, Ditmanson Medical Foundation, Chiayi Christian Hospital, Chia-Yi 60002, Taiwan; 01729@cych.org.tw; 3Department of Nephrology, Department of Internal Medicine, Chi Mei Medical Center, Yongkang District, Tainan 710, Taiwan; 4Department of Biological Science and Technology, Chung Hwa University of Medical Technology, Rende District, Tainan City 717, Taiwan; 5Department of Urology, Tainan Municipal An-Nan Hospital, Tainan 709, Taiwan; 6Department of Urology, National Cheng Kung University Hospital, College of Medicine, National Cheng Kung University, Tainan 704, Taiwan

**Keywords:** kidney sparing surgery, upper tract urothelial carcinoma, endoscopy, prognosis

## Abstract

We compared the outcomes in early-stage upper tract urothelial carcinoma (UTUC) patients receiving endoscopic ablation (EA) with radical nephroureterectomy (RNU). From 2004 to 2018, cTa/T1N0M0 UTUC patients undergoing EA and RNU were enrolled. For reducing observational bias, propensity scores based on inverse probability of treatment weighting (IPTW) were utilized for comparing the oncologic outcomes and renal function changes. In total, 65 of 184 cTa/T1 UTUC patients were analyzed after exclusion of 119 patients with end-stage renal disease, and lack of previous ureteroscopic biopsy. The studied patients included 23 who received EA and 42 RNU, and both groups were well balanced after adjusting with IPTW. The median follow-up period was 43.6 months. There was no statistical difference between the two groups in terms of oncological outcome. The EA group exhibited less estimated glomerular filtration rate (eGFR) decline one year later (0.0% vs. 20.2%, *p* < 0.001) and less worsening of chronic kidney status (13.2% vs. 46.5%, *p* = 0.026). Among patients receiving EA, high-grade tumors exhibited higher subsequent recurrence in the residual urinary tract than did the low-grade ones. (*p* = 0.037). In summary, endoscopic ablation preserves renal function without compromising oncological outcome in selected UTUC patients. High-grade tumors should be strictly followed up following endoscopic ablation.

## 1. Introduction

Upper tract urothelial carcinoma (UTUC) is an uncommon genitourinary malignancy and accounts for 5–10% of all urothelial malignancies in western countries [1]. In contrast, there is a much higher incidence of UTUC in Taiwan and the age-standardized incidence rate of UTUC per 100,000 population was 4.09 in men and 4.37 in women, respectively [2]. There were several known etiologies associated with UTUC, including chronic arsenic exposure [3], Chinese herb nephropathy [4], or Balkan nephropathy [4]. The latter two were reported to be associated with Aristolochic acid exposure and chronic kidney disease [4]. Currently, the standard therapy for localized UTUC is radical nephroureterectomy and bladder cuff resection. Endoscopic management can be considered in those with low-grade and low-stage disease [5,6,7]. Adjuvant or salvage systemic chemotherapy should be considered for those with muscle-invasive, at least, or node-positive disease [8]. Nevertheless, some patients or urologists may hesitate in undergoing nephroureterectomy because of the higher incidence of subsequent deterioration of renal function following the loss of one renal unit, particularly for those with potential progression to end-stage renal diseases (i.e., diabetics, patients with chronic kidney disease, patients with Aristolochic acid exposure), or not benefit those scheduled for post-operative chemotherapy. Some selective patients might be treated with kidney-sparing surgeries (i.e., endoscopic ablation, segmental resection of the ureter, or distal ureterectomy), if the lesion is a solitary, low-grade tumor, particularly with a pedunculated stalk [9]. Recently, the progress of immune checkpoint inhibitor has been introduced into the treatment of non-muscle invasive urothelial carcinoma [10]. Mitomycin gel has been approved for the treatment of low-grade UTUC via instillation into the ureter or renal pelvis [11]. Like chemotherapy in the neoadjuvant setting, immunotherapy or intraureteral mitomycin gel instillation can produce about 40% of the complete pathological response in selective urothelial carcinoma [11]. Taken together, it is necessary to undertake a reappraisal of the role of endoscopic ablation as a definite therapy for UTUC in the current circumstances, particularly for those high-grade, non-invasive patients.

In contrast with radical nephroureterectomy (RNU), endoscopic ablation is theoretically supposed to be able to prevent the solitary kidney status/anephric state, resulting in higher rates of subsequent dialysis, cardiovascular morbidity, and overall mortality [12,13,14,15]. Besides, endoscopic ablation may result in higher local recurrence and distant metastases if there is a lack of meticulous selection of candidates and stringent follow-up. Nevertheless, there is not enough evidence or control to support these viewpoints because the rarity of UTUC mostly precludes double blinded, placebo-controlled studies. The safety of endoscopic management as compared to RNU is mostly based on retrospective uncontrolled studies [7]. In the current study, we demonstrated that endoscopic ablation can preserve renal function without compromising oncological outcome in selected non-invasive, both low- and high-grade, UTUC patients, which was strengthened by using an inverse probability of treatment weighting analysis.

## 2. Materials and Methods

### 2.1. Patient Populations

This retrospective study was conducted after obtaining the approval from the Institutional Review Board of the National Cheng Kung University Hospital (A-ER-103-036 and A-ER-103-012). From April 2004 to February 2018, a total of 184 patients with cTa/T1N0M0 UTUC were diagnosed and treated at our hospital. Those with end stage renal disease (ESRD), without previous ureteroscopic biopsy, and without subsequent definite surgery were excluded from the study. For the purposes of comparison, other kidney sparing surgeries were also excluded, such as segmental resection of ureter, or distal ureterectomy. All the enrolled patients were thoroughly evaluated, including a standard diagnostic ureteroscopy, computed tomography (CT), excretory urography or retrograde pyelography, and urine cytologic analysis. The tumor was categorized according to the 2007 TNM staging and the 2004 WHO grade system. All the patients were treated and followed according to the treatment consensus of urothelial carcinoma of bladder and upper urinary tract, modified from the NCCN Clinical Practice Guidelines in Oncology. Our selection criteria for definitive endoscopic management of UTUC include: no evidence of parenchymal invasion on CT imaging; complete ureterorenoscopic visualization; complete endoscopic ablation; and willingness for a stringent, follow-up protocol.

All the patients were counseled on the risk and benefits of endoscopic excision and, also, of nephroureterectomy. Endoscopic approaches including semirigid ureteroscopic tumor ablation, and percutaneous nephrostomy tumor ablations were applied to these patients. Tumors were first biopsied then treated with fulguration, the Neodymium:YAG laser and/or the Holmium:YAG laser afterwards. Patients were treated and then followed on a stringent postoperative endoscopic and image protocol.

### 2.2. Statistical Analysis

Both of the oncologic outcomes and changes of renal function were compared between UTUC patients receiving kidney-sparing surgery or RNU. In order to control the potential selection bias associated with non-randomization, we performed propensity score analysis by implementing an inverse probability of treatment weighting (IPTW) method. Propensity scores were generated using generalized boosted modelling (GBM) logistic regression. Based on available evidence, preoperative baseline characteristics were selected to calculate propensity scores. Age, gender, biopsy grade, clinical stage, previous history of urothelial carcinoma, carcinoma in situ status, tumor size (≥3 cm), and hydronephrosis status were thought as the important measurable variables affecting treatment selection. The IPTW method balances the covariate of the two groups by weighting all patients of the database by the inverse of their propensity score (1/(ps)) in the endoscopic ablation group and 1/(1−ps) in the RNU group. The Kaplan–Meier method and stratified Cox proportional hazards model were used to estimate and compare overall and progression free survival. Bladder tumor recurrence-free survival (RFS) is calculated from the time of EA or RNU to the time of bladder tumor recurrence. Progression-free survival (PFS) or overall survival (OS) is calculated from the time of surgery to the time of tumor upstaging or disease-related death. Independent samples *t*-test was used to determine relationships between continuous variables. Pearson’s Chi-square test was used to evaluate categorical variables. The log rank test and the Kaplan–Meier method were applied for the univariate analysis. Multivariate Cox proportional hazards model was used to calculate predictive values of independent relationships between categorical variables that were prognostic in the univariate analysis. Analyses were calculated using SPSS version 23.0 (SPSS Inc., Chicago, IL, USA). All *p* values were two-sided and *p* < 0.05 was considered statistically significant.

## 3. Results

### 3.1. Patients’ Characteristics 

There were 119 patients excluded from this study, including those with muscle-invasive disease, ESRD, receiving other types of KSS, without ureteroscopic biopsy, and without subsequent definite therapy. A total of 65 patients were enrolled for analysis. Both the demographics and clinical characteristics of the patients are shown in Table 1. In this cohort, 23 patients (mean age, 66.0 years; 47.8% male) and 42 patients received EA and RNU (mean age, 69.3 years; 45.2% male), respectively. Based on initial endoscopic biopsy, there were 17 (73%) and 26 (60%) high-grade tumors in EA and RNU groups, respectively; 14 (60.9%) cTa, 9 (39.1%) cT1 and 31 (73.8%) cTa, 11 (26.2%) cT1 in EA and RNU groups, respectively. There were 13 (56.5%) and 7 (16.2%) patients with concomitant or previous UC in the EA and RNU groups, respectively. In the RNU group, there were upgrading (6 (14.3%) low, 36 (85.7%) high-grade tumors) and upstaging (18 (42.9%) pTa, 24 (57.1%) pT1 tumors) noted on final pathology. Except for these characteristics, it was not statistically different between these two groups in the aspects of the existing hydronephrosis, tumor size, adjuvant intravesical chemotherapy, multifocality, and preoperative renal function. 

The median preoperative serum creatinine level and estimated glomerular filtration rate were 1.35 mg/dL, 76.0 mL/min/1.73 m^2^, and 1.02 mg/dL, 61.9 mL/min/1.73 m^2^ in the EA and RNU groups, respectively. There were no statistical differences between these two groups (*p* values, 0.796, and 0.932, respectively). After IPTW adjustment, all standardized differences of weighted comparisons of all covariates were less than 5%, indicating that the distribution of baseline patients and tumor characteristics were similar between two groups preoperatively. 

### 3.2. Oncological Outcomes

Until April 2019, the patients received a median follow-up of 33.6 months (IQR, 20, 60 months). Among 23 patients with endoscopic ablation, there 14 (60.8%) patients with cancer progression, 7 (30.4%) with bladder recurrence, 1 (4.3%) with distant metastasis. Nine patients (39.1%) underwent radical nephroureterectomy due to ipsilateral or contralateral recurrence. Among 42 patients with RNU, there 17 (40.4%) patients with cancer progression, 14 (40.4%) with bladder recurrence, 4 (9.5%) with distant metastasis (4.5%). A total of 14 (33.3%) patients underwent contralateral radical nephroureterectomy. 

The patients’ outcomes in regards of 5-year OS, 5-year PFS, 5-year bladder recurrence -free survival, and changes of renal function outcomes of both groups are listed in Table 2. Kaplan–Meier analysis demonstrated that there were no differences in terms of OS (5-year OS, 94.5% vs. 94.6%, *p* = 0.987), PFS (5-year PFS, 58.6% vs. 55.8%, *p* = 0.883), and bladder cancer RFS (5-year bladder cancer RFS, 75.2% vs. 55.8%, *p* = 0.250) between endoscopic ablation and RNU cohorts (Table 2). 

With univariate and multi-variate analyses, the type of definite surgery (EA vs. RNU) was not an independent prognostic factor for overall survival or PFS (Table 3). In terms of bladder cancer recurrence-free survival, both types of definite surgery (endoscopic ablation vs. RNU, hazard ratio, 0.49; 95% CI, 0.25–0.97; *p* = 0.042) and tumor multiplicity (single vs. multiple, hazard ratio, 4.17; 95% CI, 1.32–13.2; *p* = 0. 015) were significant prognostic factors in univariate analysis; however, only tumor multiplicity was an independent poor prognostic factor for subsequent bladder recurrence (single vs. multiple, hazard ratio, 3.62, 95% CI, 1.01–13.0; *p* = 0. 049) (Table 3). Among 23 patients receiving EA, there were no differences between low- and high-grade tumors, except for subsequent urinary tract recurrence. High-grade tumors exhibited higher tumor recurrence in the urinary tract (log-rank test, *p* = 0.037), as well as bladder tumor recurrence (log-rank test, *p* = 0.074). (Table 4) (Figure 1A,B).

### 3.3. Outcomes of Renal Function

One year after surgery, the estimated glomerular filtration rate (eGFR) declined 0.0% and 20.2% in the EA and RNU groups, respectively (*p* < 0.001). According to the definition of the National Kidney Foundation, 12 (52%) and 18 (42%) patients were categorized as CKD stage 3 at the time of definite surgery; one year later, 16 (69%) and 35 (83%) patients in the EA and RNU groups, respectively. The RNU cohort exhibited a higher percentage of CKD worsening (more than stage 3) as compared with the EA cohort. (46.5% vs. 13.2%, *p* = 0.026) (Table 2).

After weighting, univariate analysis of linear regression for eGFR changes after surgery showed that type of definite surgery (EA vs. RNU, β ± SE, 17.9 ± 4.13; *p* < 0.001), gender (male vs. female, β ± SE, −16.1 ± 4.33; *p* < 0.001), and tumor multiplicity (single vs. multiple, β ± SE, −19.96 ± 4.69; *p* < 0.0001) were significant factors for subsequent eGFR worsening. Multivariable linear regression analysis showed that both type of definite surgery (EA vs. RNU, β ± SE, 12.9 ± 4.92; *p* =0.011) and gender (male vs. female, β ± SE, −12.89 ± 4.02; *p* = 0.002) were independent predictive factors (Table 5). 

## 4. Discussion

Using propensity score weighting, we demonstrated that early stage UTUC patients receiving endoscopic ablation can diminish the worsening of renal function without sacrificing oncological control, as compared with those receiving nephroureterectomy in the current study. Since the incidence of UTUC is low, it is difficult to perform a randomized trial or avoid any potential selection bias. Until now, the published studies discussing this concept were usually small case series, or lacked comparable groups. Although the present study was a retrospective, non-randomized, non-controlled trial, we verified this concept using the Inverse Probability Treatment Weighting (IPTW) propensity score method for reducing the effect of the selection bias. 

The oncological outcome from EA treatment in our study was compatible with those in the published studies [16,17]. Keeley FX, et al. reported a 5-year PFS rate of 64% in a cohort of UTUC received EA [12]. Adam S et al. reported a 5-year OS rate of 86–93% in a cohort of UTUC received EA [13]. The results of these EA treatments were not different from the results of patients receiving RNU in the current study. Regardless of treatment modality, there were 15–50% subsequent bladder tumor recurrences following the definite surgery for UTUC [14]. In addition, the majority of bladder tumor recurrences will occur within the first two years of initial definite surgery for UTUC. Our result demonstrated that high-grade tumors receiving EA exhibited a higher frequency of subsequent UC recurrence, including bladder tumor recurrence. Overall, RNU and tumor multiplicity were positively associated with subsequent bladder cancer recurrence in univariate analysis and only tumor multiplicity is an independent factor. The reasons may be intertwined. Actually, it is thought that the EA group would have more recurrences, owing to the intact ureter and renal pelvis, not only this factor, but also the fact that there were many factors influencing the subsequent bladder recurrence, including molecular subtype, tumor staging and grading, previous history of bladder UC, and intravesical chemotherapy. There were more frequencies of a single tumor in the patients from the EA cohort. The patients of the EA cohort usually receive more strict ureterorenoscopic surveillance under spinal or intravenous general anesthesia, rather than local anesthesia. In contrast, the patients of the RNU cohort receive cystoscopic surveillance mainly under local anesthesia. Despite this, the renal salvage rate was 60.86% as compared with those of RNU cohort. There were nine recurrences which led to radical nephroureterectomy (eight ipsilateral and one at contra lateral kidney) in our EA cohort. Taken together, the EA treatment can reduce the rate of renal unit loss without sacrificing the oncological overall survival in selected UTUC patients. However, the surveillance for tumor recurrence in the residual urinary tract should be strictly performed in the high-grade tumors. 

In the current study, linear regression multivariate analyses for eGFR changes demonstrated both RNU and male gender factors were independent indicators for worsening renal function in early stage UTUC patients. In term of the RNU factor, this finding just proved the concept. In addition, we found a 13.2% deterioration of eGFR change in the EA cohort. There were several reasons contributing to deterioration of renal function in the EA subgroup, including regular ureteroscopic follow-up, Double-J stenting, intravesical chemotherapy, disease recurrence, frequent surveillance of either ureterorenoscopic ablation intervention, or upper urinary tract imaging. Moreover, it is difficult to understand why the male factor can influence the renal function. The reason may require further investigation. 

UTUC was an uncommon disease in western countries and it is becoming more prevalent owing to the exposure to Aristolochic acid-containing Chinese traditional medicine or Balkan cereals. So, it is important to maintain renal function in treating patients with such etiologies, harboring chronic kidney disease. Our recent study demonstrated that more than 90 percent of the studied tumor specimens exhibited DNA adduct induced by Aristolochic acid exposure, regardless of superimposed chronic arsenic exposure or not [18]. In the current study, we utilized the method “inverse probability of treatment weighting” (IPTW) to diminish the effects of observed confounding, because it is very difficult to conduct a full-matched study for the proof of the concept. 

There were several limitations in the current study. First, our study design is retrospective and with a small number of cases. With the IPTW propensity score method, the selection bias can be diminished. Second, the instrument used in the endoscopic ablation procedure was inconsistent, either electrocauterization or Nd-YAG or Holmium LASER, which may influence the oncological outcome or renal function. Third, it was a relatively short time for the follow-up period to observe the long-term result. Since some innovative therapies have been introduced into the therapies for UTUC, such as check point inhibitors, intraureteral instillation of mitomycin-C gel, it is necessary to collect more patients and observe for longer periods in future. Fourth, we did not discuss the impact of concomitant comorbidities on survival or functional outcome. We excluded those patients with ESRD, and the RNU group may exhibit a lower probability of comorbidities than the EA group. With this effort, the impact of concomitant comorbidities may not produce as much inferiority in the RNU group. Fifth, clinical staging is well known to be inaccurate in UTUC which may lead to bias despite IPTW. Therefore, we recruited those with stage Ta and T1 in both clinical and pathological staging into the RNU group to reduce the bias to survival.

## 5. Conclusions

We demonstrated in the current study that endoscopic management of urothelial carcinoma is a feasible approach. For clinical tumor stage Ta/T1 UTUC, endoscopic ablation provides equivalent oncological outcomes and minimizing of renal function deterioration in comparison with RNU. For those high-grade tumors, strict surveillance following endoscopic ablation is required. Since its rarity, it is hard to provide full-matching evidence for this observation; our study used IPTW for diminishing the observational confounding, which provides an example for studying such uncommon diseases in future.

## Figures and Tables

**Figure 1 jcm-11-01307-f001:**
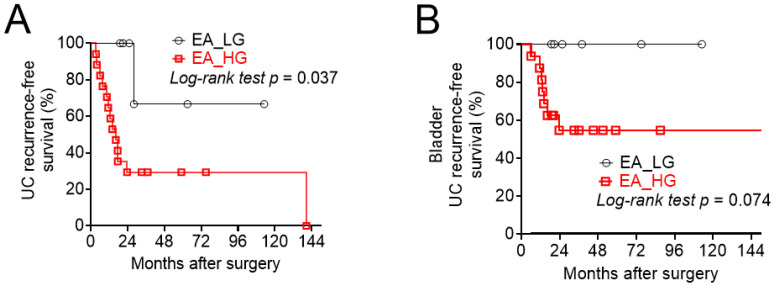
Kaplan–Meier curve for tumor recurrence in patients receiving endoscopic ablation (EA) according to tumor grade. (**A**) UC recurrence-free survival; (**B**) Bladder tumor recurrence-free survival.

**Table 1 jcm-11-01307-t001:** Basic characteristics of the study’s subjects and after weighting.

Patients Characteristics	Original Cohort		Standardized Difference	Weighted Cohort		Standardized Difference
	EA	RNU		EA	RNU	
N	23	42		23	42	
Age (years)						
median IQR	66.0 (59.0, 77.0)	69.3 (58.3, 75.6)	0.06	62.0 (30.0, 71.0)	66.9 (57.7, 74.9)	−0.26
Male, (%)	47.83	45.24	0.05	35.26	46.41	−0.23
Biopsy tumor stage, (%)						
cTa	60.87	73.81	0.28	75.49	74.06	−0.03
cT1	39.13	26.19		24.51	25.94	
Biopsy tumor grade, (%)						
Low	26.09	40.48	0.31	51.63	39.30	−0.25
High	73.91	59.52		48.37	60.70	
Pathological stage, (%)						
pTa	-	42.86	-	-	45.98	-
pT1	-	57.14		-	54.02	
Pathological grade, (%)						
Low	-	14.29	-	-	18.03	-
High	-	85.71		-	81.97	
Hydronephrosis, (%)	30.43	78.57	−1.10	53.33	69.10	−0.33
CIS, (%)						
0	100.00	69.05	−0.95	0.00	24.21	−0.80
1	0.00	30.95		100.00	75.79	
Tumor Size ≥ 3 cm, (%)	18.18 ^a^	35.71	−0.40	43.66 ^a^	34.47	0.19
Pre-operation creatinine median IQR	1.35 (0.99, 2.34)	1.02 (0.87, 1.57)	0.42	0.95 (0.53, 1.65)	0.99 (0.87, 1.56)	−0.16
Pre-operation eGFR median IQR	54.5 (27.0, 72.0)	62.4 (40.6, 82.1)	−0.45	76.0 (38.0, 90.0)	61.9 (42.3, 82.0)	0.06
Pre-operative CKD > stage 3, (%)	52.17	42.86	0.19	34.68	42.33	−0.16
Adjuvant IVCT, (%)	8.70	2.38	0.28	6.79	4.90	0.08
Previous/Conc. UC, (%)	56.52	16.67	0.91	31.82	18.58	0.31
Tumor site, (%)						
Ureter alone	56.52	40.48	−0.33	74.82	37.82	−0.80
Renal pelvis c/w Ureter	43.48	59.52		25.18	62.18	
Multifocal, (%)	73.91	100.00	−0.84	43.43	100.00	−1.61

^a^ One missing. CIS, carcinoma in situ; eGFR, estimated glomerular filtration rate; CKD, chronic kidney disease; IVCT, intravesical chemotherapy; Conc., concomitant; UC, urothelial carcinoma; c/w, with/without; IQR, interquartile range; EA, endoscopic ablation; RNU, radical nephroureterectomy.

**Table 2 jcm-11-01307-t002:** Comparison of patients’ outcomes between endoscopic ablation and radical nephroureterectomy.

Outcome	Weighted Cohort [%/Median (IQR)]		*p*
	EA	RNU	
5-year OS	94.5	94.6	0.987
5-year PFS	58.6	55.8	0.883
5-year bladder cancer RFS	75.2	55.8	0.250
Creatinine increasing after operation	77.2	94.4	0.131
eGFR changes after operation	0.0 (−11.0, 0.0)	−20.2 (−40.9, −13.0)	<0.001
CKD status worse	13.2	46.5	0.026

eGFR, estimated glomerular filtration rate; CKD, chronic kidney disease; IQR, interquartile range; EA, endoscopic ablation; RNU, radical nephroureterectomy; OS, overall survival; PFS, progression-free survival; RFS, recurrence-free survival.

**Table 3 jcm-11-01307-t003:** Univariate and multivariate Cox regression for patients’ survival.

	OS		PFS		RFS	
Variables	HR ^a^ (95% CI)	*p*	HR ^a^ (95% CI)	*p*	HR ^a^ (95% CI)	*p*
**Univariate analysis**						
EA vs. RNU	1.79 (0.35–9.13)	0.486	1.09 (0.60–1.98)	0.768	0.49 (0.25–0.97)	0.042
Age (per 5 years)	1.42 (0.92–2.20)	0.112	1.12 (1.01–1.24)	0.037	1.06 (0.96–1.18)	0.237
Male vs. Female	0.61 (0.11–3.38)	0.568	2.23 (1.24–4.03)	0.008	1.61 (0.84–3.07)	0.151
Tumor grade (High vs. Low)	2.24 (0.27–18.4)	0.452	1.86 (0.99–3.49)	0.053	1.54 (0.78–3.02)	0.211
cT1 vs. cTa	3.53 (0.65–19.1)	0.143	1.47 (0.78–2.75)	0.230	1.40 (0.69–2.80)	0.350
Previous/Conc. UC (Yes vs. No)	0.43 (0.05–3.51)	0.432	1.49 (0.81–2.74)	0.203	1.08 (0.53–2.22)	0.825
Tumor Size (≥3 vs. 3 cm)	0.17 (0.01–3.75)	0.259 ^b^	0.49 (0.25–0.95)	0.034	0.52 (0.26–1.07)	0.075
Tumor location						
(RP c/w Ureter vs. Ureter alone)	1.84 (0.31–11.1)	0.505	1.67 (0.92–3.02)	0.090	1.66 (0.87–3.17)	0.128
Hydronephrosis (Yes vs. No)	0.21 (0.03–1.68)	0.139	0.91 (0.51–1.65)	0.763	0.96 (0.50–1.84)	0.897
Adjuvant IVCT (Yes vs. No)	0.79 (0.03–18.4)	0.883 ^b^	3.40 (1.43–8.10)	0.006	2.09 (0.78–5.56)	0.142
CIS (Yes vs. No)	0.42 (0.02–9.64)	0.586 ^b^	1.44 (0.62–3.31)	0.395	2.07 (0.88–4.84)	0.095
Multifocal (Yes vs. No)	0.24 (0.01–6.14)	0.386 ^b^	2.21 (0.91–5.37)	0.080	4.17 (1.32–13.2)	0.015
**Multivariate analysis**						
EA vs. RNU					0.82 (0.38–1.75)	0.607
Age (per 5 years)			1.10 (0.96–1.26)	0.188		
Male vs. Female			2.30 (1.21–4.40)	0.012		
Tumor Size (≥3 vs. <3 cm)			0.36 (0.16–0.83)	0.016		
Adjuvant IVCT (Yes vs. No)			4.08 (1.43–11.7)	0.009		
Multifocal (Yes vs. No)					3.62 (1.01–13.0)	0.049

EA, endoscopic ablation; RNU, radical nephroureterectomy; Conc., concomitant; UC, urothelial carcinoma; c/w, with/without; IVCT, intravesical chemotherapy; CIS, carcinoma in situ; HR, hazard ratio; CI, confidence interval. ^a^ Weighted by inverse probability of treatment weights method. ^b^ Firth regression; OS, overall survival; PFS, progression-free survival; RFS, recurrence-free survival.

**Table 4 jcm-11-01307-t004:** Comparison of patients’ outcomes receiving endoscopic ablation according to tumor grade.

	Low Grade	High Grade	*p* Value
N	6	17	
Age, mean ± SD (yr)	63.2 ± 18.0	69.0 ± 10.4	0.343
Gender (M/F)	2/4	9/8	0.640
Tumor location (RP c/w Ureter/Ureter alone)	1/5	8/9	0.340
cT1 vs. cTa	0/6	8/9	0.058
Previous/Conc. UC (Yes/No)	4/2	9/8	0.660
Tumor Size (≥3/<3 cm)	1/5	3/13	0.999
Adjuvant IVCT (Yes/No)	1/5	1/16	0.463
CIS (Yes/No)	0/6	0/17	1.00
Bladder tumor recurrence (Yes/No)	0/6	8/9	0.058
Any urinary tract recurrence	1/5	13/4	0.018
Subsequent RNU (Yes/No)	1/5	8/9	0.340
Progression (Yes/No)	1/5	3/14	0.999
Death (Yes/No)	1/5	3/14	0.999

SD, standard deviation; M, male; F, female; RP, renal pelvis; Conc., concomitant; UC, urothelial carcinoma; c/w, with/without; IVCT, intravesical chemotherapy; CIS, carcinoma in situ; RNU, radical nephroureterectomy.

**Table 5 jcm-11-01307-t005:** Linear regression for eGFR changes after surgery.

Variables	Univariate Model		Multivariable Model ^b^	
	β ± SE ^a^ (95% CI)	*p*	β ± SE ^a^ (95% CI)	*p*
EA vs. RNU	17.9 ± 4.13	<0.001	12.9 ± 4.92	0.011
Age (per 5 years)	−1.12 ± 0.68	0.105		
Male vs. Female	−16.1 ± 4.33	<0.001	−12.89 ± 4.02	0.002
High grade vs. Low grade	−4.10 ± 4.71	0.387		
cT1 vs. cTa	−8.16 ± 5.32	0.130		
Previous/Conc. UC (Yes/No)	−3.39 ± 5.44	0.535		
Tumor Size (≥3/<3 cm)	8.96 ± 4.70	0.061		
Tumor location (RP c/w Ureter/Ureter alone)	−6.11 ± 4.67	0.196		
Hydronephrosis (Yes vs. No)	4.53 ± 4.81	0.350		
Adjuvant IVCT (Yes vs. No)	1.00 ± 10.09	0.922		
CIS (Yes vs. No)	−8.82 ± 6.96	0.210		
Multifocal (Yes vs. No)	−19.96 ± 4.69	<0.001	−6.34 ± 5.79	0.278

eGFR, estimated glomerular filtration rate; EA, endoscopic ablation; RNU, radical nephroureterectomy; Conc., concomitant; UC, urothelial carcinoma; c/w, with/without; IVCT, intravesical chemotherapy; CIS, carcinoma in situ. ^a^ Weighted by inverse probability of treatment; ^b^ Multivariable linear regression analysis of variables (Group variable and *p* < 0.05 in univariate linear regression).

## Data Availability

The clinical information of the studied subjects was retrieved from electronic medical records of University Hospital of NCKU. These data are not public and only available after obtaining the IRB consent.

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
