# Peer review of "Outcome of Non-Muscle Invasive Upper Tract Urothelial Carcinoma Receiving Endoscopic Ablation: An Inverse Probability of Treatment Weighting Analysis"

_jcm, 2022, doi:10.3390/jcm11051307_

Round 1

Reviewer 1 Report

The authors presented a propesity score matched comparison of functional and oncological outcomes between endoscopic ablation and nephrouretectomy for upper tract urothelial cancer.

The paper is nice and well written.

Just few minor comments:

  • concerrning the rate of patients experiencing newly onset of CKD>3 after 1 year of follow up in endoscopic ablation cohort.  What are the causes of renal function deterioration? Ureteral strictures? medical conditions?
  • No mention about comorbidities that can impact on survival and functional outcomes. These could be reported and compared as clinical features of the two study cohorts.

Author Response

Reviewer 1

The authors presented a propensity score matched comparison of functional and oncological outcomes between endoscopic ablation and nephrouretectomy for upper tract urothelial cancer.

The paper is nice and well written.

  • Thanks for your comment

Just few minor comments:

  • concerning the rate of patients experiencing newly onset of CKD>3 after 1 year of follow up in endoscopic ablation cohort.  What are the causes of renal function deterioration? Ureteral strictures? medical conditions?

à There were several reasons contributing to deterioration of renal function in the EA subgroup, including regular ureteroscopic followup, Double-J stenting, and intravesical chemotherapy, disease recurrence, frequent surveillance of either ureterorenoscopic ablation intervention or upper urinary tract imaging.

This statement will be added in the section of discussion. (Lines 274-277)

  • No mention about comorbidities that can impact on survival and functional outcomes. These could be reported and compared as clinical features of the two study cohorts.

àIndeed, concomitant comorbidities may impact on survival and functional outcomes. We have excluded those with end stage renal disease before conducting analysis. The impact of other comorbidities was not discussed in the current studies owing to small number of patients in the EA group. Actually, the RNU group may exhibit lower probability of co-morbities than the EA group. So that we added this statement into the limitation “Fourth, we did not discuss the impact of concomitant comorbidties on survival or functional outcome, Actually, we excluded those patients with ESRD, and the RNU group may exhibit lower probability of comorbidities than the EA group. With this effort, the impact of concomitnant comorbidities may not produce much inferiority in the RNU group” (Lines 297-301)

Reviewer 2 Report

Upper urothelial cancer is difficult to diagnose. The prognosis depends on the exhaustive control of the tumor. The article is interesting and the number of patients is sufficient.

Author Response

Reviewer 2

Comments and Suggestions for Authors

Upper urothelial cancer is difficult to diagnose. The prognosis depends on the exhaustive control of the tumor. The article is interesting and the number of patients is sufficient.
à Thanks for your comment.

Reviewer 3 Report

In this manuscript, the authors perform a retrospective analysis of patients from a single institution with upper tract urothelial carcinoma (UTUC) comparing initial management by endoscopic ablation (EA) vs. the more traditional radical nephroureterctomy (RNU).  They use statistical weighting techniques to compensate for the retrospective nature of the study.  They find that renal function is preserved with EA (no surprise), but importantly oncologic outcomes are not diminished compared with RNU.

This paper addresses a clinically important issue that confronts urologists world-wide. Urologists would like to do everything possible to preserve renal function, such as avoiding nephrectomy, as long as oncologic outcomes are not diminished.

The manuscript has many weaknesses, most of which are acknowledged by the authors. The most important weakness is the retrospective nature of the study. The authors are correct that prospective studies in this area have not been done and would be difficult due to the rarity of the disease. They compensate for the non-randomized nature of the study by performing inverse probability of treatment weighting (IPTW).

Despite this statistical maneuver, this paper is unlikely to be practice-changing, but it does provide some support for attempts to preserve kidneys in UTUC.

It is concerning that more than half (119 of 184) patients were excluded from the study. This surely introduces bias.

More information should be given about the success of ureteroscopic biopsy. In many cases, the samples are quite small and histopathology cannot be done.  Were these patients included or excluded?

Clinical staging is notoriously inaccurate in UTUC. This may lead to uncontrolled bias despite IPTW.

Some of the paragraphs in the Results section could be re-written by a native English scientific writer to make the data more understandable.

It seems that EA tended to have a higher recurrence-free survival compared with RNU (fewer recurrences). This seems counter-intuitive and suggests some statistical fluke. Intuitively, one would guess that EA would have more recurrences since the kidney remains in place.

It is difficult to understand the table with the multivariable analysis.  Data seems to be missing for Overall Survival.

Author Response

Reviewer 3

In this manuscript, the authors perform a retrospective analysis of patients from a single institution with upper tract urothelial carcinoma (UTUC) comparing initial management by endoscopic ablation (EA) vs. the more traditional radical nephroureterctomy (RNU).  They use statistical weighting techniques to compensate for the retrospective nature of the study.  They find that renal function is preserved with EA (no surprise), but importantly oncologic outcomes are not diminished compared with RNU.

This paper addresses a clinically important issue that confronts urologists world-wide. Urologists would like to do everything possible to preserve renal function, such as avoiding nephrectomy, as long as oncologic outcomes are not diminished.

The manuscript has many weaknesses, most of which are acknowledged by the authors. The most important weakness is the retrospective nature of the study. The authors are correct that prospective studies in this area have not been done and would be difficult due to the rarity of the disease. They compensate for the non-randomized nature of the study by performing inverse probability of treatment weighting (IPTW).

Despite this statistical maneuver, this paper is unlikely to be practice-changing, but it does provide some support for attempts to preserve kidneys in UTUC.

It is concerning that more than half (119 of 184) patients were excluded from the study. This surely introduces bias.

  • Thanks for your comments.

More information should be given about the success of ureteroscopic biopsy. In many cases, the samples are quite small and histopathology cannot be done. Were these patients included or excluded?

àIn this current study, only patients with ureteroscopic biopsy were enrolled into the study for comparison. Therefore we could not provide the successful rate of ureteroscopic biopsy. Those patients without ureteroscopic biopsy-proven UC were excluded.(Lines 86, 135)

Clinical staging is notoriously inaccurate in UTUC. This may lead to uncontrolled bias despite IPTW.

  • Thanks for your comments. We will add this statement into the limitation section. “Fifth, clinical staging is well-known inaccurate in UTUC which may lead to bias despite IPTW. Therefore, we recruit those with stage Ta and T1 in both clinical and pathological staging into the RNU group to reduce the bias on survival.” (Lines 301-303)

Some of the paragraphs in the Results section could be re-written by a native English scientific writer to make the data more understandable.

àWe had re-written some of paragraphs in the Result section. (Lines 134-226)

It seems that EA tended to have a higher recurrence-free survival compared with RNU (fewer recurrences). This seems counter-intuitive and suggests some statistical fluke. Intuitively, one would guess that EA would have more recurrences since the kidney remains in place.

àActually, it is thought that the EA group wound have more recurrence owing to the intact ureter and renal pelvis. Not only this factor, but also there were many factors influencing the subsequent bladder recurrence, including molecular subtype, tumor staging and grading, previous history of bladder UC, and intravesical chemotherapy. Thus, we added this statement in the discussion. (Lines 255-259)

It is difficult to understand the table with the multivariable analysis. Data seems to be missing for Overall Survival.

àAfter initial calculating with univariate analysis, only significant factors were used for subsequent multivariate analysis. Also, the number of selected factors for multivariate analysis follows the one-tenth of censored events. Therefore, we did not perform multivariate analysis in the aspect of overall survival. (Lines 178)